# Phytohormone Abscisic Acid Improves Memory Impairment and Reduces Neuroinflammation in 5xFAD Mice by Upregulation of LanC-Like Protein 2

**DOI:** 10.3390/ijms21228425

**Published:** 2020-11-10

**Authors:** Seung Ho Jeon, Namkwon Kim, Yeon-Joo Ju, Min Sung Gee, Danbi Lee, Jong Kil Lee

**Affiliations:** 1Department of Fundamental Pharmaceutical Science, Graduate School, Kyung Hee University, 26, Kyungheedae-ro, Dongdaemun-gu, Seoul 02447, Korea; bawoojang@khu.ac.kr (S.H.J.); yeonj00@khu.ac.kr (Y.-J.J.); 2017315113@khu.ac.kr (M.S.G.); 2Department of Life and Nanopharmaceutical Sciences, Graduate School, Kyung Hee University, 26, Kyungheedae-ro, Dongdaemun-gu, Seoul 02447, Korea; kop03@khu.ac.kr (N.K.); dannnnnnb@khu.ac.kr (D.L.); 3Department of Pharmacy, College of Pharmacy, Kyung Hee University, 26, Kyungheedae-ro, Dongdaemun-gu, Seoul 02447, Korea

**Keywords:** Alzheimer’s disease, amyloid beta, abscisic acid, LanC-like protein 2, neuroinflammation

## Abstract

Alzheimer’s disease (AD), a type of dementia, is the most common neurodegenerative disease in the elderly. Neuroinflammation caused by deposition of amyloid β (Aβ) is one of the most important pathological causes in AD. The isoprenoid phytohormone abscisic acid (ABA) has recently been found in mammals and was shown to be an endogenous hormone, acting in stress conditions. Although ABA has been associated with anti-inflammatory effects and reduced cognitive impairment in several studies, the mechanisms of ABA in AD has not been ascertained clearly. To investigate the clearance of Aβ and anti-inflammatory effects of ABA, we used quantitative real-time polymerase chain reaction and immunoassay. ABA treatment inhibited Aβ deposition and neuroinflammation, thus resulting in improvement of memory impairment in 5xFAD mice. Interestingly, these effects were not associated with activation of peroxisome proliferator-activated receptor gamma, well known as a molecular target of ABA, but related with modulation of the LanC-like protein 2 (LANCL2), known as a receptor of ABA. Taken together, our results indicate that ABA reduced Aβ deposition, neuroinflammation, and memory impairment, which is the most characteristic pathology of AD, via the upregulation of LANCL2. These data suggest that ABA might be a candidate for therapeutics for AD treatment.

## 1. Introduction

Alzheimer’s disease (AD) is one of the most common types of dementia in the elderly, but the main etiology remains unknown [1,2]. Pathologically, AD reveals excessive deposition of extracellular amyloid β (Aβ) and accumulation of neurofibrillary tangles, which are hyper-phosphorylated intracellular tau proteins [3,4,5]. The clinical symptoms of AD include cognitive disorders, memory impairment, and other behavioral indications [6]. AD is classified into two types: early onset AD (EOAD, <5%), which occurs before age 65; and late-onset AD (LOAD, >95%), which mostly occurs after age 65. The majority of cases are late-onset AD, and are linked to neuroinflammation, oxidative stress, and mitochondrial dysfunction through interrupted energy homeostasis in the brain [7,8,9,10,11].

Aβ is one of the best-known pathological hallmarks of AD and is generated by sequential cleavages of amyloid precursor protein (APP) and a number of enzymes that cleave different points, including β-secretase (BACE1) and γ-secretase. Although the exact pathology of AD is unclear, a number of studies have shown the relationship between Aβ plaques and cognitive deficits [12,13]. Many studies, including clinical research, have speculated that neurotoxicity by accumulated Aβ plaque leads to neurodegeneration in AD [14,15,16]. Neuroinflammation caused by deposition of Aβ plaque is also known to be associated with AD pathogenesis. Accumulating evidence suggests that neuroinflammation is exacerbated by activated microglia and astrocytes [17,18]. Therefore, normalization of reactive gliosis could be one of the therapeutic strategies for AD.

Abscisic acid (ABA) is an isoprenoid phytohormone to play crucial roles in numerous physiological processes in plants [19]. ABA has recently received some attention because of its therapeutic potential. Administration of ABA ameliorates inflammatory response via inhibiting cellular adhesion molecule expression and immune cell infiltration in dextran sodium-sulfate-induced colitis [20]. Sánchez-Pérez AM proposed that regular intake of ABA can prevent memory dysfunction via anti-inflammatory effects [21]. Another study reported that ABA attenuates cognitive impairment induced by streptozotocin in rats [22]. It has not been clearly established how ABA works on AD. Thus, the aim of this study is to investigate the therapeutic potential of ABA and related mechanisms in the pathology of AD.

## 2. Results

### 2.1. ABA Improved Cognitive Competence in 5xFAD Mice

To determine the effect of ABA on cognitive function, Morris water maze (MWM) was performed with five familiar Alzheimer’s disease (5xFAD) mice widely used in AD animal models. Five-months-old wild type (WT) and 5xFAD mice littermates were divided into 4 groups: vehicle-treated WT, ABA-treated WT, vehicle-treated 5xFAD, ABA-treated 5xFAD, and treated vehicle or ABA for 2 months. Cognitive competence was determined by the time to arrive at the hidden platform. On the first day, the record of reaching the platform was similar in all groups. Compared with the vehicle-treated WT group, the vehicle-treated 5xFAD group showed significantly longer escape latency and the gap between two groups was statistically significant since day 6. However, ABA-treated 5xFAD mice showed decreasing escape latency compared to vehicle-treated 5xFAD mice since day 4, and the escape pattern was similar to the vehicle-treated WT mice (Figure 1A,B). On day 8, the platform was removed and the probe task was performed. There were no differences in the total distances traveled and average swim speeds among the four groups (Figure 1C,D). The time spent in the target quadrant and number of crossings across the platform zone were significantly decreased in the vehicle-treated 5xFAD mice compared to vehicle-treated WT mice, but these were recovered in the ABA-treated 5xFAD mice (Figure 1E,F). These results indicated that ABA improves cognitive impairment in 5xFAD mice.

### 2.2. ABA Diminished Aβ Deposition in the Brains of 5xFAD Mice

Aβ deposition causes a common and typical lesion in AD patients. To evaluate whether memory improvements by the ABA administration is related to Aβ pathology, we analyzed Aβ deposition using Thioflavin S (ThS) staining and 6E10 immunostaining. Compared with the vehicle-treated 5xFAD group, ThS-positive plaque was significantly decreased in hippocampus of ABA-treated 5xFAD mice (Figure 2A,B). The level of Aβ deposition was significantly reduced in the ABA-treated 5xFAD mice brain compared with vehicle-treated 5xFAD mice (Figure 2C,D). Then, we measured the protein level of Aβ in the brain by using a human specific Aβ_40_ and Aβ_42_ enzyme-linked immunosorbent assay (ELISA) kit. ELISA results showed that the levels of Aβ_40_ and Aβ_42_ in both the cortex and hippocampus of the 5xFAD mice were reduced by ABA treatment (Figure 2E,F). These results showed that ABA treatment could ameliorate memory dysfunction via decreasing Aβ deposits in the brain of 5xFAD mice.

### 2.3. ABA Reduced Hippocampal C99 Levels in the 5xFAD Mice Brains

Since changes in APP processing or degradation by ABA could lead to a reduction of Aβ levels in the ABA-treated 5xFAD mice brain, we investigate whether ABA mediates APP processing and degradation through protein levels. The protein levels of APP and presenilin1 (PS1) involved in Aβ production were markedly increased in the brain of vehicle-treated 5xFAD mice compared with that of vehicle-treated WT mice (Figure 3A–C,G–I). There were no significant differences in the expressions of APP and PS1 between the vehicle-treated 5xFAD and ABA-treated 5xFAD group in both the cortex (Figure 3A–C) and hippocampus (Figure 3G–I), indicating that ABA does not affect Aβ processing. There were also no significant differences between the groups in the expression levels of insulin degrading enzyme (IDE) and neprilysin (NEP), which are involved in Aβ degradation (Figure 3A,D,E,G,J,K). However, the expression of C99, a β-secretase-derived βAPP fragment, was decreased in the cortex and hippocampus of the ABA-treated 5xFAD mice relative to the expression in the vehicle-treated 5xFAD mice (Figure 3A,F,G,I). These data represented that although Aβ was reduced by ABA treatment, ABA does not affect Aβ production and degrading enzymes.

### 2.4. ABA Decreased Inflammatory Responses in the 5xFAD Mice Brains

Inflammatory mediators enhance APP processing and induce Aβ formation. Activated astrocytes and microglia also worsen the brain through release of inflammatory factors [23]. To date, several studies have reported that ABA have anti-inflammatory effects [24,25]. We investigated whether ABA mediates inflammatory factors in mice brain. To identify the astrocyte, we stained glial fibrillary acidic protein (GFAP) as astrocyte marker. In immunofluorescence analysis of mice brain tissue, the expression of GFAP was increased in the vehicle-treated 5xFAD mice relative to that in WT mice. However, the expression of GFAP was significantly reduced in the ABA-treated 5xFAD mice compared to the vehicle-treated 5xFAD mice (Figure 4A,B). We also stained ionized calcium-binding adaptor molecule 1 (Iba-1) as microglia marker. Similar to GFAP results, Iba-1-positive area was significantly reduced in 5xFAD mice after ABA (Figure 4C,D). Next, we analyzed mRNA expression levels of proinflammatory cytokines, such as tumor necrosis factor (TNF-α), interleukin-1β (IL-1β), and interferon-γ (IFN-γ) in the cortex and hippocampus. The mRNA expression of TNF-α, IL-1β, and IFN-γ were downregulated in ABA-treated 5xFAD mice relative to those in vehicle-treated 5xFAD mice (Figure 4E,F). These results indicated that treatment of ABA could moderate inflammatory response in 5xFAD mice brain.

### 2.5. ABA Did Not Affect the PPAR-γ Pathway, but Did Increase LANCL2-Mediated CREB Expression

ABA has been reported to have anti-inflammatory effects through peroxisome proliferator-activated receptor gamma (PPAR-γ) activity in RAW 264.7 macrophage [25]. Several studies have suggested that PPAR-γ regulates inflammatory responses and PPAR-γ agonists were potential therapeutic treatment for AD [26]. To examine the positive effect of ABA associated with the PPAR-γ pathway, we assessed the expressions of PPAR-γ and peroxisome proliferator-activated receptor gamma coactivator 1-alpha (PGC-1α). There were no significant differences in the protein and mRNA levels of PPAR-γ and PGC-1α in the mice brains among all groups (Figure 5). Another possible mechanism leading to downregulation of proinflammatory factor by ABA is through the LanC-like protein 2 (LANCL2)-mediated effects. ABA is known to be a natural ligand of LANCL2 [27,28,29]. We confirmed that ABA treatment could increases the protein level of LANCL2 in the cortex and hippocampus of the ABA-treated 5xFAD mice relative to that in the vehicle-treated 5xFAD mice (Figure 6). We also observed that cAMP response element-binding protein (CREB), a downstream factor of LANCL2, was increased in the brain of 5xFAD mice (Figure 6). These results suggested that ABA might affects LANCL2-mediated CREB expression.

## 3. Discussion

AD is initiated by deposition of Aβ plaques and tau tangles and causes severe neuroinflammation. In clinical studies, brain specimens of AD patients showed Aβ plaques, and many studies tried to inhibit Aβ associated factors [30,31]. Neuroinflammation caused by aggregated Aβ is a well-known AD lesion type [18]. Several experiments have verified the effect of ABA in inflammation-related diseases, such as inflammatory bowel disease (IBD) and diabetes. Moreover, in a high-fat diet-induced model, ABA has been shown to ameliorate neuroinflammation [32,33,34,35]. Some studies have proven that ABA prevented neuroinflammation and cognitive impairment; however, the related mechanism was not elucidated [22,36]. In this study, we demonstrated that the positive effects of ABA on AD pathologies are associated with anti-Aβ and anti-inflammation effects in 5xFAD mouse model.

Interruption of APP processing and inhibition of Aβ aggregation are the main therapeutic strategies for AD [37]. We examined Aβ deposition in the cortex and hippocampus of the 5xFAD mice to inspect the positive effects of ABA. Compared with vehicle-treated 5xFAD mice, treatment of ABA for 2 months reduced the plaque areas and protein levels of Aβ in both the cortex and hippocampus. We also found decreasing Aβ levels by ELISA. We investigated APP generation and degradation-related proteins, including APP, PS1, IDE, and NEP, by using western blot analysis. These protein levels were not changed in the ABA-treated 5xFAD group relative to those in the vehicle-treated 5xFAD group, indicating that ABA did not affect Aβ generation and degradation. The C-terminal fragment of APP, C99, is generated by BACE1 [38]. We measured the BACE1 protein levels, but no changes were observed (data not shown). However, the protein level of C99 was reduced in the ABA-treated 5xFAD mice relative to the level in the vehicle-treated 5xFAD mice brains. Possible hypothesis is that ABA is concerned in the α-secretase, which interfere with the formation of C99. Further study on whether ABA might increase levels of α-secretase and disrupt the structure of C99 is needed.

Inflammatory factors contributing to the AD pathology is well known. Activated microglial cells secrete proinflammatory cytokines, such as TNF-α, IL-1β, and IL-6 [39]. These cytokines from microglia are concerned with neuroinflammation that accelerates the AD pathologies including Aβ accumulation [40]. Impairment of function including the water channel protein aquaporin-4 in astrocyte also exacerbates Aβ pathologies [41,42,43,44]. Both are known to contribute to Aβ production and accumulation [45]. Modulation of inflammatory conditions by regulating microglia and astrocyte attenuates memory impairment and Aβ accumulation [46]. We showed that ABA decreased inflammatory factors, such as TNF-α and IL-1β via inhibiting gliosis in 5xFAD mice. Reduction of C99, a β-secretase cleaved protein, might be due to the delay in APP processing by downregulated inflammatory factors. These results suggest that ABA delayed Aβ accumulation through anti-inflammatory effects.

PPAR-γ agonists have been shown to decrease the number of activated microglia and astrocytes and attenuate Aβ deposits in both the cortex and hippocampus [47]. Since ABA is known as PPAR-γ agonists [25], we investigated whether positive effects of ABA in AD are associated with the expression of PPAR-γ and coactivator PGC-1α. Interestingly, in our experiment, no changes were observed in both the mRNA expressions and protein levels of PPAR-γ and PGC-1α in the cortex and hippocampus. We then investigated another ABA binding protein, LANCL2. LANCL2 is highly expressed in the brain and has recently emerged as a therapeutic target for inflammatory diseases and diabetes [48]. It was also reported that LANCL2 agonist ameliorated IBD by regulation of T-cell stability through activation of LANCL2 and PKA/CREB pathways [49]. The PKA/CREB pathways activated by LANCL2 are independent of PPAR-γ and are significant signaling pathways engaged in the regulation of inflammatory cytokines [48,50]. The protein levels of LANCL2 were upregulated in the ABA-treated 5xFAD mice compared to those in the vehicle-treated 5xFAD mice in the cortex and hippocampus. The levels of CREB in the cortex and hippocampus were increased in the ABA-treated 5xFAD group relative to those in the vehicle-treated 5xFAD group. Collectively, our data suggested that ABA might affect Aβ pathology through LANCL2, not PPAR-γ. However, it is necessary to confirm using LANCL2 knock-out mice whether the interference of LANCL2 is related to development of AD. Recently, the effort of disease modeling and drug screening for AD has continued [51,52]. Using brain organoid or brain-on-a-chip devices for drug screening that specifically bind to LANCL2 could be a way to uncover the relationship between AD and LANCL2.

In conclusion, our results showed therapeutic effects of ABA in a 5xFAD mouse model of AD. We confirmed that ABA reduced Aβ deposition in the 5xFAD mice brains and, therefore, ameliorated cognitive competence. Moreover, neuroinflammatory factors involved in the AD pathologies were reduced in the brains of ABA-treated 5xFAD mice. Our study showed that ABA did not activate PPAR-γ, but affects the LANCL2. Given that activated LANCL2 regulates inflammatory factors, activation of LANCL2 by ABA downregulates inflammatory cytokines, such as TNF-α, IL-1β, and IFN-γ. We observed that ABA reduced Aβ deposition and excessive neuroinflammation through LANCL2 activation. Overall, this study suggests that ABA might effectively delay the Aβ-mediated AD progression through downregulation of inflammatory factors. 

## 4. Materials and Methods 

### 4.1. Materials

ABA (A0792) was purchased from Tokyo Chemical Industry, ThS (T1892), anhydrous dimethyl sulfoxide (DMSO), triton X-100, paraformaldehyde (PFA), phosphate buffer (PB), phosphate-buffered saline (PBS, pH 7.4), and tris-buffered saline (TBS) were purchased from Sigma-Aldrich (St. Louis, MO, USA). RIPA buffer (89901) and protease/phosphatase inhibitor cocktail (78445) were obtained from Thermo Fisher Scientific (Waltham, MA, USA). Fluorescence mounting medium (S3023) was purchased from Dako (Santa Clara, CA, USA). Primary antibodies against Aβ (6E10; 803001, BioLegend, San Diego, CA, USA), PGC-1α (sc-518025, Santa Cruz Biotechnology, Dallas, TX, USA), PS1 (5643s, Cell Signaling Technology, Danvers, MA, USA), PPAR-γ (2443s, Cell Signaling Technology), p-CREB (9198s, Cell Signaling Technology), CREB (9197s, Cell Signaling Technology), IDE (ab32216, Abcam, Cambridge, MA, USA), LANCL2 (MBS154355, MyBioSource, San Diego, CA, USA), GFAP (Z0334, Dako), Iba-1 (019-19741, Wako Chemical, Osaka, Japan), and NEP (AF1126, R&D Systems, Minneapolis, MN, USA) were used. β-actin (sc-47778HRP, Santa Cruz Biotechnology) and horseradish peroxidase (HRP)-conjugated secondary antibodies were purchased from Santa Cruz Biotechnology. Alexa Fluor secondary antibodies were purchased from Thermo Fisher Scientific.

### 4.2. Animals and Treatment

5xFAD mice were purchased from Jackson Laboratory (stock number: 034840-JAX, Bar Harbor, ME, USA). Mice were housed in plastic cages under constant temperature (23 ± 1 °C) and humidity (50 ± 10%), in a 12-h light/dark cycle with free access to food and water. 5xFAD transgenic mice overexpress human APP695 with three familial mutations (Swedish [K670N, M671L], Florida [I716V], and London [V717I]) and human PS1 with two familial mutations (M146L and L286V) under the control of the murine Thy1 promotor. Five-month-old 5xFAD female (22 ± 2 g) and male (30 ± 2 g) mice were genotyped and divided into 4 groups: vehicle-treated WT, ABA-treated WT, vehicle-treated 5xFAD, ABA-treated 5xFAD. According to the previous report [53], mice were injected intraperitoneally with 20 mg/kg of ABA in saline (0.9% NaCl) per day for 2 months. All animal studies were performed in accordance with the “Principles of Laboratory Animal Care” (National Institutes of Health publication number 80-23, revised 1996) and approved by the Animal Care and Use Guidelines Committee of Kyung Hee University (approval number: KHUASP(SE)-17-126-1, approval date: 6 November 2018).

### 4.3. Morris Water Maze

MWM task was used to evaluate spatial memory performance. The MWM task was executed as described in our previous study [54]. Briefly, the water maze was performed using mice that had been adapted to the maze one day before training. A platform was assigned to each mouse and the submerged platform in the maze was set at a fixed position throughout the training session. The animals were placed on different starting points in each of the three trials. In a room composed of cues on different sides, the mouse swam 60 s to find the platform. The training session is composed a series of three trials per day for seven successive days. The time to mount the platform was recorded as latency for each trial and the average of value was used for data. On day eight, the platform was removed and a probe test was performed. Each mouse was placed in the quadrant of the pool on the opposite side to where the platform was located and allowed to swim for 60 s. All trials were recorded and analyzed by the free tracking software tool Toxtrac. The software, user manual, and documentation are available at https://toxtrac.sourceforge.io. The operators responsible for experimental procedure and data analysis were blinded and unaware of group allocation throughout the experiments.

### 4.4. Brain Tissue Preparation

Female mice were euthanized after behavioral testing by administration of a mixture of ketamine and xylazine in saline (0.9% NaCl) as the anesthetic, and cardiac perfusion was performed immediately using 4% PFA in 0.1 M PB. After perfusion, brains were removed, post-fixed by PFA overnight at 4 °C, and incubated in 30% sucrose at 4 °C until equilibration. Sequential 25 μm-thick coronal sections were prepared by a cryostat (CM1850; Leica, Wetzlar, Germany) and stored at −20 °C.

### 4.5. Thioflavin S Staining

Free-floating sections were incubated for 10 min in 1% thioflavin S dissolved in 50% ethanol, followed by two washes with 50% ethanol for 5 min each and one wash with distilled water for 5 min; the sections were then mounted using mounting medium.

### 4.6. Immunofluorescence

Free-floating sections were incubated for 1 h in PBS containing 3% normal goat serum, 1% bovine serum albumin (BSA), and 0.4% triton X-100. In the same buffer solution, the sections were then incubated for 24 h in primary antibodies at 4 °C. The following antibodies were used: anti-6E10, Iba-1, and GFAP. For visualization, the sections were incubated with Alexa-Fluor-488-conjugated secondary antibodies for GFAP and Alexa-Fluor-594-conjugated secondary antibodies (1:1000; Thermo Fisher Scientific, A32740) for 6E10 and Iba-1 for 1 h at room temperature. Images of the sections were captured using BX51 immunofluorescence microscope (Olympus, Tokyo, Japan). Image J software (National Institutes of Health, Bethesda, MD, USA) was used for the quantification.

### 4.7. Aβ_40_ and Aβ_42_ Enzyme-Linked Immunosorbent Assays

Aβ_40_ and Aβ_42_ ELISA assays were performed using fluorescent-based ELISA kit (Invitrogen, Camarillo, CA, USA) and appropriate Aβ standards according to the manufacturer’s protocol. In each male mouse, the hippocampus and frontal cortex from one hemisphere were homogenized in guanidine buffer containing 50 mM Tris and 5 M guanidine HCl (pH 8.0). Homogenates were mixed at room temperature for 4 h and diluted in PBS containing 5% BSA, 0.03% tween 20, and protease inhibitor cocktail (Calbiochem, San Diego, CA, USA). All Aβ standards and experimental samples were run in duplicates, and the results were averaged.

### 4.8. RNA Isolation and Quantitative Real-Time Polymerase Chain Reaction 

Quantitative real-time polymerase chain reaction (qRT-PCR) was performed to measure mRNA transcripts of cytokines. qRT-PCR was executed as described in our previous study [55]. Briefly, total RNA was extracted from the cortex and hippocampus of male mice using the Hybrid-R™ (GeneAll, Seoul, Korea), and RNA concentrations were measured using a Nanodrop ND-1000 spectrophotometer (Thermo Fisher Scientific). RNA samples (3 μg) were converted to cDNA using TOPscript™ RT DryMIX (Enzynomics, Daejeon, Korea). cDNA was amplified and quantified by qRT-PCR using TOPreal™ qPCR 2X PreMIX (SYBR Green; Enzynomics) and the CFX Connect real-time PCR system (Bio-Rad Laboratories, Hercules, CA, USA).

### 4.9. Western Blot Analysis

Western blot analysis was performed as previously described [55]. The brain tissues of male mice were blended in RIPA buffer with protease/phosphatase inhibitors by homogenizer. Equal amounts of protein samples were quantified through Bradford assay (40 μg), then diluted in sodium dodecyl sulfate (SDS) sample buffer. Samples were spilled in SDS polyacrylamide gel and separated by electrophoresis, then transferred to polyvinylidene difluoride membranes by electrophoresis. The membranes that have been transferred were reacted with blocking solution (5% BSA with 0.1% tween 20 in TBS) for 1 h at room temperature and washed three times for 5 min by washing buffer (0.1% tween 20 in TBS). Then, they are incubated with primary antibodies overnight at 4 °C. The membranes were rinsed three times with washing buffer for 5 min each and reacted with horseradish-peroxidase-conjugated secondary antibodies for 1 h. The membranes were rinsed three times for 10 min each with washing buffer. Protein detection was performed using an ECL reagent (Bio-Rad Laboratories) and captured by Solo6S EDGE (Vilber, France). The intensity of bands was quantified by Image J software.

### 4.10. Statistical Analysis

All data were expressed as the mean ± standard error of the mean (SEM). using Graph Pad Prism 5.0 software (Graph Pad software Inc., San Diego, CA, USA). Comparisons between two groups were performed with Student’s t test. Most results were analyzed statistically by one-way analysis of variance (ANOVA) followed by Tukey’s post-hoc test. Comparisons between groups over times in the MWM were analyzed by two-way ANOVA with Bonferroni post-hoc test. A value of *p* < 0.05 was considered statistically significant compared with each group. The operators responsible for experimental procedure and data analysis were blinded and unaware of group allocation throughout the experiments. No significant difference was found between male and female mice in behavioral test; thus, data obtained from male and female mice were combined. To minimize gender differences, male mice were used for protein and mRNA analysis and female mice were used for immunohistochemistry.

## Figures and Tables

**Figure 1 ijms-21-08425-f001:**
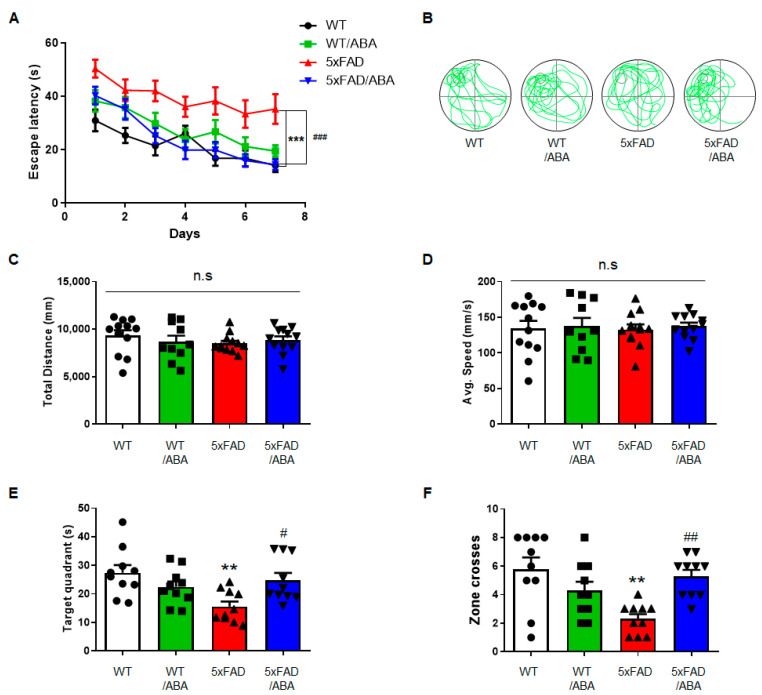
Abscisic acid (ABA) improved cognitive competence in five familiar Alzheimer’s disease (5xFAD) mice. (**A**) The Morris Water Maze (MWM) was performed to examine spatial learning and memory. The time to reach the platform was measured continually over 7 days. In a probe test (the platform was removed), (**B**) representative swimming track, (**C**) total distance, (**D**) average speed, (**E**) target quadrant, and (**F**) zone crosses were measured (*n* = 10–12 per group). All results are expressed as the mean ± standard error of the mean (S.E.M). The data were analyzed by (**C**–**F**) one-way analysis of variance (ANOVA) with Tukey’s post hoc test and (**A**) two-way ANOVA with Bonferroni *post-test*. ** *p* < 0.01, *** *p* < 0.001, significantly different from the vehicle-treated WT group; ^#^
*p* < 0.05, ^##^
*p* < 0.01, ^###^
*p* < 0.001, significantly different from the vehicle-treated 5xFAD group; n.s, non-significant.

**Figure 2 ijms-21-08425-f002:**
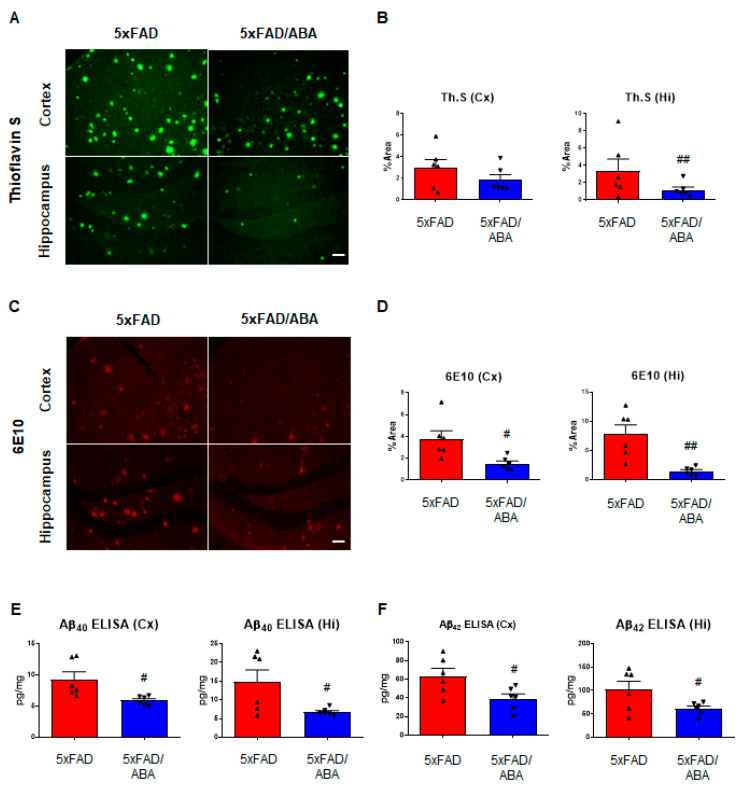
ABA reduces amyloid-beta (Aβ) deposition in 5xFAD mice brains. (**A**) Representative images of Thioflavin S (ThS) staining (scale bar = 50 μm). (**B**) Quantification of ThS positive area in the cortex and hippocampus of ABA-treated or nontreated 5xFAD mice. (**C**) Representative images of 6E10 staining, which detects the Aβ_1–16_ peptides (scale bar = 50 μm). (**D**) Quantification of 6E10 positive areas in the cortex and hippocampus of the ABA-treated or nontreated 5xFAD mice. (**E**,**F**) Results of enzyme-linked immunosorbent assay (ELISA). Levels of (**E**) Aβ_40_ and (**F**) Aβ_42_ in the cortex and hippocampus of 5xFAD mice brains. All data are expressed as the mean ± S.E.M. (*n* = 6). Statistics were analyzed by Student’s *t* test. ^#^
*p* < 0.05, ^##^
*p* < 0.01, significantly different from the vehicle-treated 5xFAD group.

**Figure 3 ijms-21-08425-f003:**
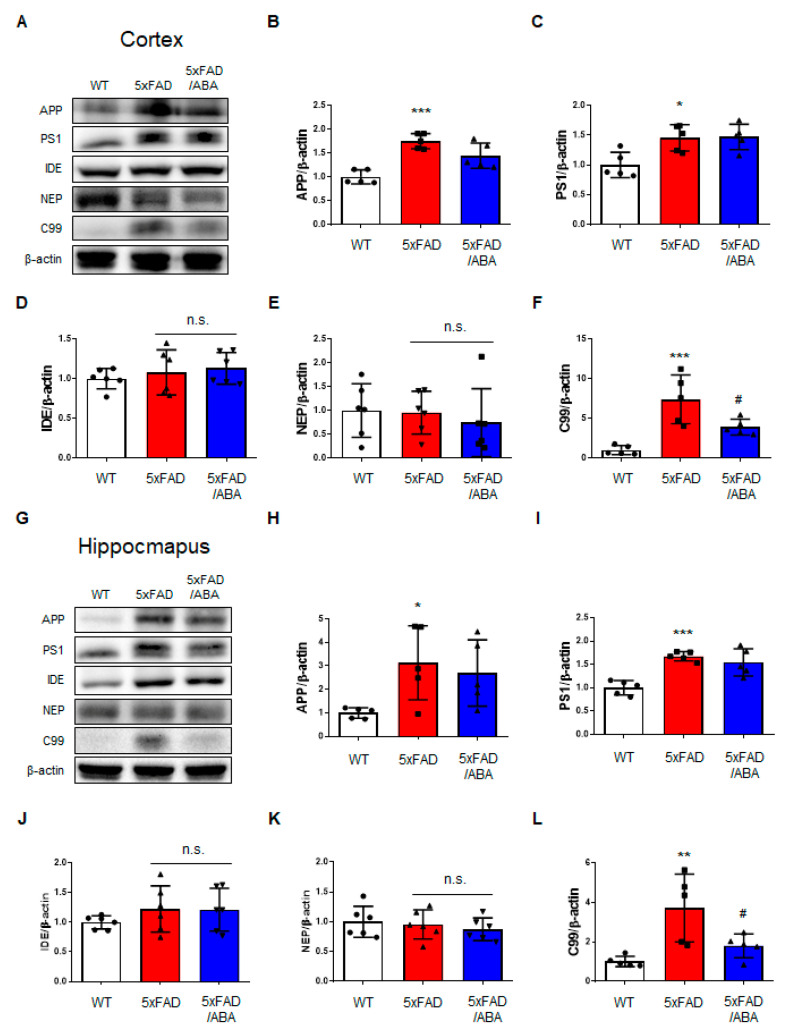
ABA decreases the C-terminal fragment 99 (C99) levels in the 5xFAD mice brains. (**A**) Representative images of immunoblotting and quantitative protein levels for (**B**) amyloid precursor protein (APP), (**C**) presenilin1 (PS1), (**D**) insulin-degrading enzyme (IDE), (**E**) neprilysin (NEP), and (**F**) C99 in the cortex of the mice brain. (**G**) Representative images of immunoblotting and quantitative protein levels for (**H**) APP, (**I**) PS1, (**J**) IDE, (**K**) NEP, and (**L**) C99 in the hippocampus of mice brain. All data are expressed as the mean ± S.E.M. (*n* = 5–6). Statistics were analyzed by one-way ANOVA with Tukey’s post hoc test. * *p* < 0.05, ** *p* < 0.01, *** *p* < 0.001, significantly different from the vehicle-treated WT group; ^#^
*p* < 0.05, significantly different from the vehicle-treated 5xFAD group; n.s., non-significant.

**Figure 4 ijms-21-08425-f004:**
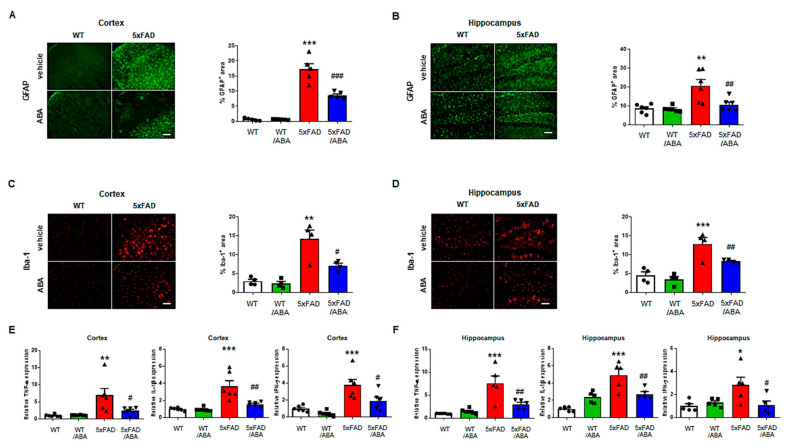
ABA ameliorates inflammation related factors in the 5xFAD mice brains. (**A**,**B**) Representative images of glial fibrillary acidic protein (GFAP) and the percentages of GFAP positive areas in the (**A**) cortex and (**B**) hippocampus quantified by using Image J software (*n* = 5–6, scale bar = 50 μm). (**C**,**D**) Representative images of ionized calcium-binding adaptor molecule 1(Iba-1) and the percentages of Iba-1 positive areas in the (**C**) cortex and (**D**) hippocampus quantified by using Image J (*n* = 4, scale bar = 50 μm). (**E**,**F**) The expression of tumor necrosis factor alpha (TNF-α), interleukin-1β (IL-1β), and interferon-γ (IFN-γ) in the (**E**) cortex and (**F**) hippocampus (*n* = 5–6). All data are expressed as the mean ± S.E.M. Statistics were analyzed by one-way ANOVA with Tukey’s post hoc test. * *p* < 0.05, ** *p* < 0.01, *** *p* < 0.001, significantly different from the vehicle-treated WT group; ^#^
*p* < 0.05, ^##^
*p* < 0.01, ^###^
*p* < 0.001, significantly different from the vehicle-treated 5xFAD group.

**Figure 5 ijms-21-08425-f005:**
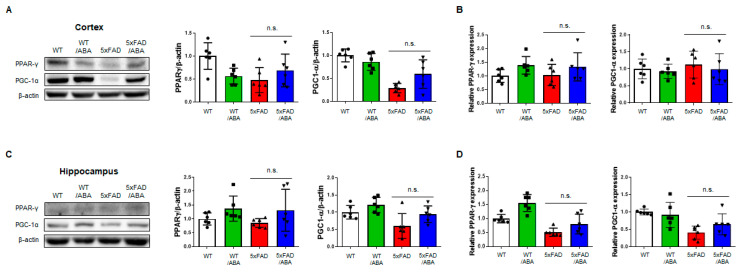
ABA does not activate the peroxisome proliferator-activated receptor gamma (PPAR-γ) pathway. (**A**) The protein levels of PPAR-γ and peroxisome proliferator-activated receptor gamma coactivator 1-alpha (PGC-1α) in the cortex. (**B**) The mRNA expressions of PPAR-γ and PGC-1α in the cortex. (**C**) The protein levels of PPAR-γ and PGC-1α in the hippocampus. (**D**) The mRNA expressions of PPAR-γ and PGC-1α in the hippocampus. All data are expressed as the mean ± S.E.M. (*n* = 6). Statistics were analyzed by one-way ANOVA with Tukey’s post hoc test; n.s., non-significant.

**Figure 6 ijms-21-08425-f006:**
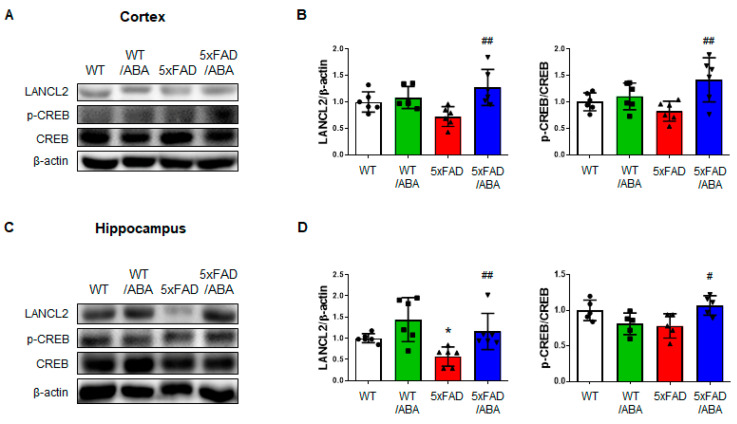
ABA attenuates AD pathologies by activating the LanC-like protein 2 (LANCL2) pathway. (**A**) The representative images and (**B**) quantified protein levels of LANCL2 and cAMP response element-binding protein (CREB) in the cortex. (**C**) The representative images and (**D**) quantified protein levels of LANCL2 and CREB in the hippocampus. All data are expressed as the mean ± S.E.M. (*n* = 5–6). Statistics were analyzed by one-way ANOVA with Tukey’s post hoc test. * *p* < 0.05, significantly different from the vehicle-treated WT group; ^#^
*p* < 0.05, ^##^
*p* < 0.01, significantly different from the vehicle-treated 5xFAD group.

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
