# Peer review of "Phytohormone Abscisic Acid Improves Memory Impairment and Reduces Neuroinflammation in 5xFAD Mice by Upregulation of LanC-Like Protein 2"

_ijms, 2020, doi:10.3390/ijms21228425_

Round 1

Reviewer 1 Report

In this manuscript, the authors show cognitive improvement after administration of ABA in 5xFAD animal model. They show how this treatment reduces amyloid deposition and inflammatory markers, and also how the treatment increases the expression of LANCL2. The experiments are well conducted and presented. 

Major comments:

  • Results section should begin with an explanation of the approach (when the drug is administered, for how long, etc) which will help readers to follow 
  • Results should be represented showing all individual points of each animal (this should be applied to the whole manuscript but it is of special importance for behavioral tests). 
  • Figure 1: Can the authors clarify when do they perform a 2-way ANOVA vs one-way ANOVA?
  • Have the authors performed any other behavioral tests? With only one they should not overstate the conclusions 
  • Can authors state if behavioral tests and imaging quantification have been performed in a blinded manner?
  • Figure 2: correct statement about Thioflavin S (it is NOT specific of amyloid beta, so they should not overstate the results)
  • Figure 3: how do the authors explain the reduction in C99? They should address in the discussion and provide some hypothesis
  • Figure 4: for accounting for neuroinflammation authors should perform more specific imaging analysis such as the morphology of microglia. ELISAs of inflammatory markers would be also informative. 
  • Figure 5 & 6: the WT/ABA group is missing and should be included
  • Figure 6C: a better image of LANCL2 should be included as it is not representative of the results provided in the qualification graph 
  • The authors show that ABA increases the expression of LANCL2 but this effect does not explain that the mechanism of action is through this pathway. To show that, authors should include an inhibitor of LANCL2 and study if in the presence of the inhibitor the protective effects are lost. If they don't perform those experiments then they must rewrite the abstract/discussion/etc. They cannot  CANNOT make the conclusion that the protective effect is through LANCL2
  • Authors have included Bothe male and female animals. Have they observed differences in terms of sex? Weather they have found differences or not is informative and could be included as supplementary material 

Minor comments: 

  • Authors must explain the acronyms the first time they appear in the text (MWM, IDE, NEP..)
  • Give short explanations of some of the approaches for better understanding (why 5xFAD model, why thioflavin S, etc). Just a short sentence is enough
  • Check grammar and typos

Author Response

We appreciated to the reviewer for sincere confirm and comments. The answer to the reviewer was attached in word file. Please see the attachment.

Reviewer 2 Report

Dear Editor

The study by Jeon et al. reports that isoprenoid phytohormone abscisic acid (ABA) reduced Aβ deposition, ameliorated cognitive competence, and inhibited neuroinflammatory factors implicated in the AD pathologies through a mechanism mediated by activation of LANCL2 but not PPAR-γ using 5xFAD mice model.

The design of the study and the technical quality of the work are convincing and results can be of general interest. The manuscript is well-written and easy to follow. Data have been presented in good way and authors used correct statistical approaches in analyzing the results.

However, there is a number of major and minor points that would need to be addressed in order to improve the quality of this paper before it can be accepted for publication:

General:

-Bar charts should be replaced with much more informative scatter plots or similar to enhance the visibility of the data and provide a more transparent insight to the distribution of the presented results. This suggestion is based on the following article:

https://onlinelibrary.wiley.com/doi/epdf/10.1111/ejn.13400

-Define abbreviations whenever they appear first in the manuscript and use them throughout. Example: MWM in line 21.

Major:

-Authors need to mention how they determine the dose of ABA in this study and whether they’ve investigated the dose-dependent effect of ABA.

-The study lacks a control when investigating the effect on ABA as PPAR agonist. Author should compare its effects to one of the known agonists such as Azelaoyl PAF or others to enable us and the readers having a direct comparison.

Minor:

-Line 41, P1: authors mentioned oxidative stress and the role of mitochondria but they need to expand this a little bit by touching upon the emerging role of energetic brain in neurodegeneration, particularly AD. References to be included:

-https://www.ncbi.nlm.nih.gov/pmc/articles/PMC5094909/

-https://pubmed.ncbi.nlm.nih.gov/31318452/

-Line 16, P2: “However, no researches establish”, this needs to be rephrased to something like it hasn’t been established …

-Line 25, P2: “decreased with the passage of time”. Authors need to be more specific with the overall duration and when the effect started to be seen.

-Line 17, P5: “GFAP, which is expressed by astrocytes under inflammatory conditions”, this statement isn’t strictly true. Astrocytes are not all necessarily GFAP positive. Authors need to have a look on the new results reported by King et al, Brain Sciences 2020 and also discuss considering other biomarkers such as AQP4. This has been nicely shown by the work of Kitchen et al Cell 2020. The parallel expression of GFAP and AQP4 might be a good marker, not only for differentiated astrocytes or reactive astrocytes, but also for precursor cells. References to be included:

- https://www.ncbi.nlm.nih.gov/pmc/articles/PMC7463428/

-https://www.ncbi.nlm.nih.gov/pmc/articles/PMC7242911/

-Line 6, P6: “The mRNA expression of TNF-α, IL-1β, and IFN-γ were ameliorated”, it might be better to mention fold changes to provide a better insight for the readers.

-Line 19-21, P6: “Several studies have suggested that oxidative stress and mitochondrial dysfunction were major contributing factors to the pathophysiology of AD and that PPAR-γ affects mitochondrial

function and inflammatory response [23, 24]”. Authors have cited some work that indicated that Wnt/β-catenin was involved in the underlying mechanism. This is an interesting point since the same mechanism has been shown before to mediate the effect of hypoxia on primary astrocytes. Authors need to expand on this point here or in the discussion since targeting this mechanism might be a therapeutic target for future studies based on the current work. References to be included:

-https://pubmed.ncbi.nlm.nih.gov/29311824/

-https://pubmed.ncbi.nlm.nih.gov/31120770/

-Line 19-20: “Impairment of astrocyte function also exacerbates Aβ pathologies”, authors need to discuss the role of AQP4 and glymphatic pathway in this mechanism. References to be included:

- https://pubmed.ncbi.nlm.nih.gov/22896675/

- https://www.ncbi.nlm.nih.gov/pmc/articles/PMC4598011/

- https://www.ncbi.nlm.nih.gov/pmc/articles/PMC7242911/

- https://pubmed.ncbi.nlm.nih.gov/30561329/

Best.

Author Response

(The authors gave the same response as above.)

Round 2

Reviewer 1 Report

The authors have addressed most of the comments and the manuscript has significantly improved. I consider that the manuscript is suitable for publication

Author Response

Thank the reviewer for careful review. We hope that the positive outcome would be educe.

Reviewer 2 Report

Dear Editor

I would like to thank the authors for their efforts to revise the manuscript in the light of the raised concerns and suggestions. The newly added details, improved figures, and updated references have helped towards the improvement of the current version compared to their earlier submission.

The majority of my comments have been addressed accordingly but the manuscript can be improved by pointing out to future directions which should be added at the end of discussion.

I would like to recommend this manuscript for publication at IJMS once the below mentioned points are duly addressed.

Best.

-Page 9 Line 11 in the discussion: authors need to briefly discuss future directions following this study which could include, but not limit to, the use of humanized 3D self-organized models, organoids and organ-on-a-chip platforms for human brain endothelial cells and BBB since the main focus of this study is to identify new targets for a human disease, and hence these elements are crucial for new therapeutic developments through ensuring efficient drug delivery. References to be included:

- https://www.ncbi.nlm.nih.gov/pmc/articles/PMC6679380/

- https://www.ncbi.nlm.nih.gov/pmc/articles/PMC7576009/

- https://pubmed.ncbi.nlm.nih.gov/30806304/

Author Response

(The authors gave the same response as above.)
